# Depressive-like Behaviors Induced by mGluR5 Reduction in 6xTg in Mouse Model of Alzheimer’s Disease

**DOI:** 10.3390/ijms241613010

**Published:** 2023-08-21

**Authors:** Youngkyo Kim, Jinho Kim, Shinwoo Kang, Keun-A Chang

**Affiliations:** 1Department of Health Science and Technology, Gachon Advanced Institute for Health Sciences & Technology, Gachon University, Incheon 21999, Republic of Korea; 2Department of Pharmacology, College of Medicine, Gachon University, Incheon 21999, Republic of Korea; 3Department of Molecular Pharmacology and Experimental Therapeutics, Mayo Clinic, Rochester, VT 55905, USA; 4Neuroscience Research Institute, Gachon University, Incheon 21565, Republic of Korea

**Keywords:** Alzheimer’s disease (AD), mGluR5, depressive-like behavior

## Abstract

Alzheimer’s disease (AD) is one representative dementia characterized by the accumulation of amyloid beta (Aβ) plaques and neurofibrillary tangles (NFTs) in the brain, resulting in cognitive decline and memory loss. AD is associated with neuropsychiatric symptoms, including major depressive disorder (MDD). Recent studies showed a reduction in mGluR5 expression in the brains of stress-induced mice models and individuals with MDD compared to controls. In our study, we identified depressive-like behavior and memory impairment in a mouse model of AD, specifically in the 6xTg model with tau and Aβ pathologies. In addition, we investigated the expression of mGluR5 in the brains of 6xTg mice using micro-positron emission tomography (micro-PET) imaging, histological analysis, and Western blot analysis, and we observed a decrease in mGluR5 levels in the brains of 6xTg mice compared to wild-type (WT) mice. Additionally, we identified alterations in the ERK/AKT/GSK-3β signaling pathway in the brains of 6xTg mice. Notably, we identified a significant negative correlation between depressive-like behavior and the protein level of mGluR5 in 6xTg mice. Additionally, we also found a significant positive correlation between depressive-like behavior and AD pathologies, including phosphorylated tau and Aβ. These findings suggested that abnormal mGluR5 expression and AD-related pathologies were involved in depressive-like behavior in the 6xTg mouse model. Further research is warranted to elucidate the underlying mechanisms and explore potential therapeutic targets in the intersection of AD and depressive-like symptoms.

## 1. Introduction

Alzheimer’s disease (AD) is the most common neurodegenerative disease and the most common neurodegenerative disease of dementia [1]. Around 6.2 million Americans aged 65 and older live with AD, a number that could reach 13.8 million by 2060 without medical advancements [2]. A major neurological feature of AD is the intracellular accumulation of Neurofibrillary tangle (NFT) containing hyper-phosphorylated tau protein due to extracellular accumulation of Aβ accompanied by memory loss and cognitive degradation [1]. Neuropsychiatric symptoms, including irritability, anxiety, and major depressive disorder (MDD), are prevalent in more than 80% of AD patients [3,4].

Metabotropic glutamic receptor subtype 5 (mGluR5), which plays a crucial role in regulating synaptic transmission, is a target associated with mood disorders [5]. The density of mGluR5 was reduced in the brains of stress-induced mice models and MDD-affected individuals [6,7,8]. Recently, evidence linking mGluR5 to AD pathophysiology, as well as MDD, has been provided [9], but changes in mGluR5 expression in the brains of AD patients and animal models remain controversial [10,11,12,13,14]. The study conducted in AβPP transgenic mice (tg-ArcSwe) using [11C]-ABP688-PET did not reveal any differences in mGluR5 changes compared to wild-type (WT) mice [10]. However, immunoblot analysis showed increased levels of mGluR5 protein in tg-ArcSwe mice [10]. In 5xFAD mice, both [18F]-FPEB-PET and immunoblot analysis demonstrated lower mGluR5 binding density and protein levels in the hippocampus and striatum compared to WT mice [11]. Another study focusing on 9-month-old 5xFAD mice reported a statistically significant decrease in mGluR5 protein levels compared to WT mice [12]. Furthermore, investigations using PET imaging with [18F]FPEB or [11C]-ABP688 in human AD brains revealed reductions in hippocampal mGluR5 binding during early AD stages compared to controls [13,14].

In addition, mGluR5 can regulate the PI3K/AKT/GSK-3β pathway in the hippocampus, and this modulation has the potential to reverse Aβ-induced neurotoxicity [15,16]. Glycogen synthase kinase-3β (GSK-3β) has emerged as a prominent kinase involved in the pathological hyper-phosphorylation of the tau protein [17,18]. Increased phosphorylation of GSK-3β at Tyr216 residue has been implicated in the hyperphosphorylation of tau, and this process is mediated through the PI3K/AKT/GSK-3β signaling pathway [16,18]. Dysfunction of the PI3K/AKT signaling pathway leads to elevated GSK-3β activity and subsequent tau hyperphosphorylation, contributing to the formation of NFTs [18]. These previous studies have provided insights into potential interactions between mGluR5 and AD; however, evidence linking mGluR5 signaling directly to AD remains limited.

Previous studies have provided evidence linking GSK-3β to the pathophysiology of MDD. Specifically, human post-mortem studies have indicated GSK-3β activity in the ventral prefrontal cortex of MDD patients compared to control subjects [19,20]. Additionally, animal models of stress-induced depression have shown increased GSK-3β activity [21]. Notably, the downregulation of AKT activity in the ventral tegmental area (VTA) has been implicated in regulating susceptibility to depressive behaviors in these models [8,22].

Based on these findings, our study aims to investigate depressive-like behavior resulting from alterations in the mGluR5 expression in the brains of an AD mouse model, specifically the 6xTg model with tau and Aβ pathologies [23]. By exploring these aspects, we sought to shed light on the potential involvement of mGluR5 and related signaling pathways in the development of depressive symptoms in the context of tau and Aβ pathologies.

## 2. Results

### 2.1. Expression of Aβ and tau Pathologies and Cognitive Impairment in 6xTg Mice

In this study, the AD pathologies and the cognitive impairment were reconfirmed in 4-month-old 6xTg mice with Aβ and NFTs, as well as memory loss in our previous study [23] (Figure 1). The expressions of human APP and human tau were confirmed in 6xTg mice but not wild-type (WT) mice (Figure 1A). In histological analysis, Aβ and p-tau were observed in the brains of 6xTg mice but not WT mice (Figure 1B).

To confirm memory impairment, behavioral experiments were performed including the Y-maze test, passive avoidance test (PAT), and Morris water maze (MWM) test. In the Y-maze test, spontaneous alteration of 6xTg mice (54.79 ± 2.90%, *** *p* < 0.001) was significantly decreased compared to WT (77.80 ± 1.70%), but the total number of arms entered was not significantly different between the groups (Figure 1C). To investigate the impact on short-term reference memory, a PAT was performed. The latency time of 6xTg (46.50 ± 7.85 s, *** *p* < 0.001) was significantly reduced compared to WT (214.1 ± 15.84 s) (Figure 1D). Finally, the spatial and working memory was evaluated using the MWM test. On the last day of acquisition training, 6xTg mice (42.33 ± 3.74 s, *** *p* < 0.001) showed significantly increased escape latency compared to WT mice (19.66 ± 6.78 s). During probe tests, 6xTg (19.84 ± 2.90 s) also showed a decrease in the latency time in the target zone compared to the WT group (28.02 ± 3.90 s) (Figure 1E). However, the velocity was not significantly different between the groups (Figure 1E). These results show that 6xTg mice have memory deficiency caused by Aβ and tau pathologies. A previous in vitro study showed that high levels of Aβ, especially in its oligomeric form, altered glutamatergic synaptic transmission and caused synapse loss [24], and synaptic loss was also demonstrated in 6xTg mice (Appendix A).

### 2.2. Depressive and Anxiety-like Behaviors of 6xTg Mice

Next, depression-like behavior was investigated in 6xTg mice compared to littermate WT mice using Forced Swim Test (FST) and Novelty Suppressed Feeding (NSF) tests (Figure 2A and Appendix A). Although depressive-like behaviors including FST and NSF in 4-month-old mice were not significant differences between the groups (Appendix A), 6- and 8-month-old 6xTg mice (6 M, 342.9 ± 7.92 s, ** *p* < 0.01, 8M, 344.1 ± 21.27 s, ** *p* < 0.01) showed a significant increase in immobility time compared to WT mice (6 M, 187.3 ± 37.85 s, 8 M, 144.3 ± 44.74 s) (Figure 2B). In the NSF test, the latency time of 6- and 8-month-old 6xTg mice (6 M, 164.3 ± 33.44 s, * *p* < 0.05, 8 M, 119.7 ± 19.27 s, ** *p* < 0.01) was significantly increased compared to the WT group (6 M, 78.83 ± 14.74 s, 8 M, 33.83 ± 4.09 s) (Figure 2C). However, the amount of intake in 8-month-old 6xTg mice (0.33 ± 0.02 s, * *p* < 0.05) was significantly decreased compared to littermate WT mice (0.46 ± 0.05 s) (Figure 2C).

Anxiety disorders as well as depressive symptoms are commonly observed in individuals diagnosed with MDD [25]. Then, we investigated the anxiety-like behaviors using the Elevated plus maze (EPM) and Open field test (OFT) in 4- and 6-month-old 6xTg mice compared to littermate WT mice (Appendix A). In EPM, latency time of closed arm in 6xTg mice increased from 4 months (6xTg; 38.91 ± 6.43, * *p* < 0.05 vs. WT; 23.39 ± 2.45) to 6 months (6xTg; 48.76 ± 5.12, ** *p* < 0.01 vs. WT; 29.89 ± 3.33), and frequency of closed arm in 6xTg mice increased from 4 months (6xTg; 29.86 ± 5.873, ns vs. WT; 16.50 ± 2.432) to 6 months (6xTg; 26.57 ± 2.458, ** *p* < 0.01 vs. WT; 12.83 ± 2.023) (Appendix A). In OFT, cumulative duration in the center zone was significantly decreased in 6 months old 6xTg (7.571 ± 0.1, ** *p* < 0.01) compared to WT mice (16.50 ± 2.63), but the distance was not significantly different between the groups (Appendix A). These results indicate that 6xTg mice, which have amyloid and tau pathologies, appeared in depression- and anxiety-like behaviors.

### 2.3. Reduced Expression of mGluR5 Protein in the Brains of 6xTg Mice

Previous studies have shown a reduction in mGluR5 density in the brains of AD patients and animal models [10,11,12,13,14] as well as MDD [7,8]. Based on this evidence, the mGluR5 expression level was investigated in the brain of 4- and 6-month-old 6xTg mice using micro-PET with [11C]-ABP688. In the brains of 6-month-old 6xTg mice, mGluR5 density was significantly decreased in the cortex (CX) (0.5 ± 0.06, ** *p* < 0.01) and hippocampus (HP) (0.49 ± 0.02, **** *p* < 0.0001) compared to littermate WT mice (CX; 1.00 ± 0.01, HP; 1.00 ± 0.02). However, no significant difference in mGluR5 density was seen between 4-month-old 6xTg and WT mice (Figure 3A–D).

To confirm the decrease in mGluR5 protein levels in 6xTg mice, Western blot analysis was performed using brain tissues obtained from the cortex and hippocampus of 2-, 4-, 6-, 8-, and 10-month-old 6xTg mice and littermate WT mice (Appendix A). The protein levels of mGluR5 were significantly reduced in the cortex and hippocampus of 6xTg mice starting at 4 months of age compared to WT mice (Appendix A). Especially, the protein levels of mGluR5 were significantly reduced in the cortex (4M; 6xTg, 0.57 ± 0.12, ** *p* < 0.01 vs. WT, 1.00 ± 0.06, 6 M; 6xTg, 0.29 ± 0.08, **** *p* < 0.0001 vs. WT, 1.00 ± 0.06) and the hippocampus (4 M; 6xTg, 0.54 ± 0.07, *** *p* < 0.001 vs. WT, 1.00 ± 0.06, 6 M; 6xTg, 0.32 ± 0.06, ** *p* < 0.01 vs. WT, 1.00 ± 0.09) of 4- and 6-month-old 6xTg mice compared to WT mice (Figure 3E,F). Furthermore, we also compared mGluR5 expression levels in the 6xTg of mice with littermate 5xFAD and JNPL3 mice, and the 6xTg mice, which have amyloid-β and tau pathologies, showed the most significant reduction in mGluR5 expression in the brain compared to the other groups (Appendix A). In the same pattern as WB, immunofluorescent staining of mGluR5 showed a reduction in mGluR5 expression in the prefrontal cortex (PFC) (6xTg, 0.88 ± 0.01, * *p* < 0.05 vs. WT, 1.00 ± 0.04), dentate gyrus (DG) (6xTg, 0.87 ± 0.04, * *p* < 0.05 vs. WT, 1.00 ± 0.04), CA1 (6xTg, 0.91 ± 0.001, ** *p* < 0.01 vs. WT, 1.00 ± 0.01), and CA3 (6xTg, 0.78 ± 0.02, *** *p* < 0.001 vs. WT, 1.00 ± 0.02) regions of 6-month-old 6xTg mice compared to the corresponding regions in the WT mice (Appendix A).

### 2.4. Alteration of ERK/AKT/GSK-3β Signaling Pathway in the Brains of 6xTg Mice

In a stress-induced mouse model, mGluR5 reduction was associated with a depressive phenotype mediated through ERK and AKT [8]. GSK-3β also plays an important role in MDD or AD brains through AKT/GSK-3β signaling pathway [4,26,27]. Therefore, we observed the downstream signaling molecules of mGluR5, such as ERK, AKT, and GSK-3β, using WB analysis. We first investigated the phosphorylated and total forms of ERK (p-ERK and t-ERK) and found a significant decrease in p-ERK/t-ERK ratio in the cortex and hippocampus of the 6xTg group compared to the WT group (CX; 6xTg; 0.56 ± 0.09, ** *p* < 0.01 vs. WT; 1.00 ± 0.07, HP; 6xTg; 0.69 ± 0.10, * *p* < 0.05 vs. WT; 1.00 ± 0.08), but there was no difference in t-ERK expression (CX; 6xTg; 1.05 ± 0.04 vs. WT; 1.00 ± 0.06, HP; 6xTg; 1.04 ± 0.05 vs. WT; 1.00 ± 0.10) (Figure 4A,B). Next, we checked the phosphorylated and total forms of AKT (p-AKT, t-AKT), and the p-AKT/t-AKT ratio was significantly reduced in the cortex and hippocampus of 6xTg mice compared to WT mice (CX; 6xTg; 0.56 ± 0.07, *** *p* < 0.001 vs. WT; 1.00 ± 0.05,), (HP; 6xTg; 0.50 ± 0.05, *** *p* < 0.001 vs. WT; 1.00 ± 0.08) (Figure 4A,B). However, there was no difference in t-AKT expression (CX; 6xTg; 0.96 ± 0.05 vs. WT; 1.00 ± 0.03), (HP; 6xTg; 0.90 ± 0.05 vs. WT; 1.00 ± 0.04) (Appendix A). Additionally, we checked the Tyr216 phosphorylated and total form of GSK-3β (p-GSK-3β(Y216), t-GSK-3β) and found a substantial increase in the p-GSK-3β(Y216)/t-GSK-3β ratio in the cortex and hippocampus of 6xTg mice when compared to WT mice (CX; 6xTg; 1.28 ± 0.05, * *p* < 0.05 vs. WT; 1.00 ± 0.10, HP; 6xTg; 1.42 ± 0.11, * *p* < 0.05 vs. WT; 1.00 ± 0.10) (Figure 4A,B). But there was no difference in t-GSK-3β expression (CX; 6xTg; 0.98 ± 0.03 vs. WT; 1.00 ± 0.02, HP; 6xTg; 1.01 ± 0.03 vs. WT; 1.01 ± 0.020) (Appendix A). The increase in the p-GSK-3β/t-GSK-3β ratio indicates elevated enzyme activity and may contribute to abnormal tau phosphorylation and amyloid production [17,18]. Therefore, we observed the amounts of phosphorylated tau (p-tau) using three kinds of antibodies (AT8 (Ser202, Thr205), AT180 (Thr231), and pS396 (Ser396) antibodies) and Aβ1-42 amounts using ELISA assay in the brains. As expected, the increased p-tau was shown in the cortex of 6xTg mice compared to WT mice (AT8; 6xTg; 2.04 ± 0.36, * *p* < 0.05 vs. WT; 1.00 ± 0.22, AT180; 2.74 ± 0.42, ** *p* < 0.01 vs. WT; 1.00 ± 0.22, pS396; 1.84 ± 0.17, * *p* < 0.05 vs. WT; 1.00 ± 0.20) (Figure 4E,F). Similar results were obtained in the hippocampus (AT8; 6xTg; 2.31 ± 0.36, ** *p* < 0.01 vs. WT; 1.00 ± 0.15, AT180; 6xTg; 2.13 ± 0.20, ** *p* < 0.01 vs. WT; 1.00 ± 0.19, pS396; 6xTg; 2.02 ± 0.24, ** *p* < 0.01 vs. WT; 1.00 ± 0.17) (Figure 4G,H). In an ELISA assay to quantify the amounts of Aβ1-42, 6xTg mice revealed a notable increase in Aβ levels in both the cortex and hippocampus of 6xTg group (CX, 158.7 ± 18.59 pg/mL, **** *p* < 0.0001, HP, 200.6 ± 0.31 pg/mL, **** *p* < 0.0001) compared to the WT group (CX, 4.06 ± 0.61, HP, 3.94 ± 0.05) (Figure 4I,J).

### 2.5. Correlation between Depressive-like Behavior and AD Pathology in 6xTg

We evaluated the correlation of immobility time in the FST, which reflects depressive symptoms, with mGluR5. We also investigated the correlation between depressive like-behavior and AD pathologies including Aβ42 or p-tau expression levels in the cortex and hippocampus of 6xTg mice. As a result, in the cortex, the correlation between depressive-like behavior and mGluR5 levels was significantly negatively correlated (FST&mGluR5 r = −0.6396, * *p* < 0.05) (Figure 5A). The depressive symptom (FST) was significantly positively correlated with the amount of Aβ42 in the cortex of the 6xTg mice (FST&Aβ42 r = 0.7018, * *p* < 0.05) (Figure 5A). In the correlation of the depressive symptom with p-tau expression levels, we found a significant positive correlation in the cortex (FST&AT8, r = 0.6212, * *p* < 0.05, FST&AT180, r = 0.1652, * *p* < 0.05) (Figure 5A). However, we found a positive, albeit non-significant, correlation between the depressive score and pS396 levels in the cortex (FST&pS396, r = 0.5789, *p* = 0.0620) (Figure 5A).

Next, we also investigated the correlation between depressive-like behavior and mGluR5 levels or AD pathologies including Aβ42 or p-tau expression levels in the hippocampus of 6xTg mice. As a result, the correlation between depressive-like behavior and mGluR5 levels was significantly negatively correlated in the hippocampus (FST&mGluR5 r = −0.6029, * *p* < 0.05) (Figure 5B). The depressive symptom (FST) was significantly positively correlated with the amount of Aβ42 in the hippocampus (FST&Aβ42 r = 0.7467, ** *p* < 0.01) (Figure 5B) of the 6xTg mice. In the correlation of the depressive symptom with p-tau expression levels, we found a significant positive correlation in the hippocampus (FST&AT8, r = 0.7867, ** *p* < 0.01, FST&AT180, r = 0.7382, ** *p* < 0.01) (Figure 5B). However, we found a positive, albeit non-significant, correlation between the depressive score and pS396 levels in the hippocampus (FST&pS396, r = 0.3730, *p* = 0.2585) (Figure 5B). These findings suggested that depressive-like behavior in the AD mice model was associated with AD pathologies as well as reduced mGluR5 expression.

## 3. Discussion

Numerous clinical studies have consistently shown that individuals with Alzheimer’s disease (AD) experience psychological symptoms such as depressive moods and anxiety [28,29,30]. Major depressive disorder (MDD) has been identified as both a risk factor for dementia and an early sign of underlying AD and other forms of dementia [31,32,33]. Notably, research has revealed that midlife depressive symptoms are associated with a 20% increase in dementia diagnosis, while late-life symptoms are linked to a 70% increase [34]. Moreover, there is a synergistic effect when anti-dementia cholinesterase inhibitors are combined with antidepressants like venlafaxine, tianeptine, and duloxetine in patients with dementia [35]. A previous preclinical study conducted on 3-month-old APP/PSEN1-Tg mice has further supported the connection between depressive- and anxiety-like behaviors, along with memory impairments [36]. Therefore, based on the available evidence, it can be concluded that MDD is a significant condition associated with AD. In this study, we investigated the relationship between psychological symptoms and AD pathologies, especially focusing on the dysregulation of mGluR5 in 6xTg mice, which have Aβ plaques and tau hyperphosphorylation as well as memory impairment [23,37]. We found increased depressive-like behaviors in 6xTg mice compared to littermate mice.

The glutamatergic system received the most attention in the pathophysiology of depression [38]. Especially, mGluR5, a key component of the glutamatergic system, is known to play a critical role in the pathophysiology in the brain of MDD [6,7,8,39] or AD [11,12,14]. In AD pathologies, the alterations in mGluR5 levels can be attributed to amyloid pathology and neuro-inflammation [12,40]. For example, the presence of amyloid plaques in the early stages of AD directly affects glutamate signaling [12]. Eventually, neurotoxic factors such as abnormal inflammatory responses induced by amyloid plaques, tangles, and activated glial cells lead to neuronal loss, resulting in a decrease in the number of available receptors on the cell membrane [12,41,42]. To investigate mGluR5 expression in the brain, [11C]-ABP688, a PET probe, is utilized in clinical and preclinical studies [14,43]. A previous study using [11C]-ABP688 PET showed lower mGluR5 levels of the brain in the depressive group, consistent with the mGluR5 expression in the post-mortem brain [7]. We also found reduced mGluR5 affinity in the brain of 6xTg mice in micro-PET imaging with [11C]-ABP688. Decreased mGluR5 protein levels were confirmed in the cortex and hippocampus of 6-month-old 6xTg mice compared to WT mice in WB analysis.

One of the G protein-coupled receptors, mGluR5, can transmit extracellular signals to the cytoplasm by activating signaling cascades and is involved in the modulation of synaptic transmission and neuronal excitability [44,45]. However, this signaling is dysregulated in pathological states such as AD [4,26,46] and MDD [8]. For instance, mGluR5 activates extracellular signal-regulated kinase (ERK) and protein kinase B (AKT) signaling pathways, and their inhibitors prevent the neuronal protective effects involved in the proliferation of progenitor cells [47]. Furthermore, AKT is known to be involved in the regulation of GSK-3β related to Aβ, p-tau protein, and the dysregulation of synaptic activity [48]. For example, activation of AKT, which consequently phosphorylates GSK-3β at Ser9, results in decreased phosphorylation of downstream substrates, such as tau and glycogen synthase. However, the phosphorylation of GSK-3β at Tyr216, which is characterized by auto-phosphorylation, is inactivated by phosphorylation of the N-terminal Ser9, resulting in increased phosphorylation of tau protein in AD [49,50,51]. These findings suggest that the reduction in mGluR5 expression could be attributed to dysregulated phosphorylation of downstream substrates such as ERK, AKT, and GSK-3β.

Our results showed decreased phosphorylation of ERK and AKT, which results in upregulated p-GSK-3β (Y216) in the cortex and hippocampus of 8-month-old 6xTg mice. This abnormal regulation of phosphorylation appears to result in neuronal cell death due to the toxicity induced by Aβ plaques and tau entanglement [52]. Moreover, we confirmed an overexpression of phosphorylated GSK-3β due to reduced p-AKT, which resulted in elevated p-tau levels in the brains of 6xTg mice. Finally, we confirmed the correlation between depressive-like behavior and AD pathologies or mGluR5 expression levels in the brain of the AD mouse model (Figure 6).

This study has some limitations. First, mGluR5 activity or expression levels must be controlled with chemical treatment or genetic regulation. Second, since only male mice were used in this study, it is necessary to confirm these findings in female AD mice. Finally, studies of specific mechanisms involving mGluR5 between AD and depressive-like symptoms are needed to develop precise targets for treatment.

In conclusion, this study suggests that reduced expression of mGluR5 may be associated with AD pathology and may contribute to the development of depressive-like behaviors in AD. These results may contribute to the development of potential therapeutic targets for psychological symptoms in AD patients.

## 4. Materials and Methods

### 4.1. Animals

We utilized 6xTg mice, which were developed in previous studies. Briefly, 6xTg were generated by crossbreeding 5xFAD transgenic mice (The Jackson Laboratory, Bar Harbor, ME, USA) with JNPL3 transgenic mice (Taconic Biosciences Inc., Germantown, NY, USA) [23]. 5xFAD involves the co-expression of the human amyloid precursor protein (APP) with the Florida (I716V), London (V717I), and Swedish (KM670N/671NL) mutations, as well as the human presenilin 1 (PS1) with the L2806V and M146L mutations. We also utilized the JNPL3 Tg mouse (Taconic Biosciences Inc., Germantown, NY, USA), which expresses human mutant tau (P301L) driven by the mouse prion promoter. Genotyping of the mice was conducted through ear cutting, DNA isolation, and polymerase chain reaction (PCR) using specific primers designed for APP, PS1, and MAPT genes. All animals were housed under controlled environmental conditions, including a temperature of 22 ± 2 °C, relative humidity of 50 ± 10%, and a 12 h light/dark cycle. They were provided ad libitum access to food and water throughout the study.

### 4.2. Behavior Test

We performed behavior tests to confirm the cognitive impairment and depression and anxiety behavior in 6xTg mice. All behavioral tests were video-recorded and analyzed using EthoVision XT9 (Noldus Information Technology, Wageningen, The Netherlands). The tests were performed in a previous study [23,53,54].

#### 4.2.1. Y-Maze

The Y-Maze was used to test memory deficits and consisted of a 3-arm structure made of white polyvinyl plastic. Each arm (A, B, and C) measured 40 cm in length, 6.8 cm in width, and 15.5 cm in height, with a folding angle of 120°. The test was conducted for 8 min, and the experiments were video-recorded. Each mouse was assigned one point when they entered the three arms in sequence. The percentage of spontaneous alternation was calculated using the formula below:$$\mathrm{Spontaneous}\;\mathrm{alternation}\;\left(\%\right)=\frac{\mathrm{Number}\;\mathrm{of}\;\mathrm{alternation}}{\left(\mathrm{Total}\;\mathrm{arm}\;\mathrm{entries}-2\right)\times 100}$$

#### 4.2.2. Passive Avoidance Test (PAT)

The passive evasion test was conducted over a span of three consecutive days using a passive evasion device (Gemini Passive Evasion System; San Diego Instruments, San Diego, CA, USA) that measures 42.5 cm in width and 35.5 cm in length. It consists of bright and dark rooms connected via a remote-operated gate. On the first day, the mice were allowed unrestricted exploration of the chambers. On the second day, as the mice entered the dark room, they received a 2 mA electric shock to their feet for a duration of 2 s, with the gate securely closed. After a 24 h interval, we recorded the latency periods of the mice, starting from when they were placed in the bright room until the moment they entered the dark chamber.

#### 4.2.3. Morris Water Maze (MWM)

Memory and spatial learning were measured using the MWM test. Mice were tested in a circular pool filled with opaque water equilibrated to room temperature (22 °C). The tank was divided into 4 quadrants with different navigation cues for each quadrant. Mice were continuously trained with four trials per mouse each day for 4 days to search for the escape platform within a maximum of 60 s. During the test, the platform location stayed constant, and the time taken to reach the platform was recorded as the escape latency. The MWM probe test was performed within 48 h of the final trial. The platform was removed from the pool, and the mice were placed in the water and allowed to swim for 60 s. The time spent in the quadrant that previously contained the platform indicates long-term memory maintenance. Swim distance, velocity, and frequency were recorded as measures of motor function.

#### 4.2.4. Open Field Test (OFT)

The OFT was conducted to measure anxiety levels in mice. The mice were put in an open field box and allowed to move freely for a duration of 10 min. Experiments were video-recorded, and the total distance was analyzed. Anxiety-like behavior was assessed by calculating the duration of time spent in the center zone as well as measuring the total distance traveled by the mice.

#### 4.2.5. Elevated Plus Maze (EPM)

The EPM measured the anxiety-like behavior of all group mice. It consisted of two closed arms (20 cm × 5 cm × 50 cm) and two open arms (20 cm × 5 cm × 50 cm). Each mouse was put in the center and approved to explore for 5 min. Experiments were video-recorded, and the total distance was analyzed.

#### 4.2.6. Forced Swim Test (FST)

The FST was conducted in two consecutive days. On the first day, each mouse was placed individually in a transparent plexiglass cylinder filled with water at a temperature of 23 ± 1 °C. The mice were allowed to swim in the water for a total of 8 min as a training session. After 24 h of the same environmental conditions, the FST was conducted in the same manner as before. Their behavior was observed and recorded for a duration of 8 min. The immobility time, which indicates depression-like behavior, was measured during the final 6 min of the test.

#### 4.2.7. Novelty-Suppressed Feeding Test (NST)

The mice were fasted for 24 h prior to the NST and had free access to 2 bottles of water during the fasting period. The floor of the open field box was covered with a 2 cm layer of wooden bedding and allowed to habituate for 30 min. The mouse’s food was placed in the center of the arena on a white circular filter paper measuring 10 cm in diameter. Each mouse was then placed in one corner of the open field box, and the latency time until the first feeding episode was recorded. Once the mouse began eating, it was moved alone to its home cage with food provided for 5 min. Afterward, the weight of the remaining food was measured to calculate the food intake [55].

### 4.3. Micro-PET Study with [11C]-ABP688

Micro-PET scanning was conducted using a Siemens Focus 120 small animal PET scanner (Siemens Preclinical Solutions Inc., Erlangen, Germany) with a 30 min list mode acquisition protocol. For the PET imaging, [11C]-labeled PET probes (7–10 MBq) were administered to the mice through the tail vein as a single bolus. Prior to the scan, the mice were anesthetized using a mixture of 2% isoflurane and 98% oxygen.

The dynamic list mode data obtained from the scan were organized into sonograms consisting of 15 frames. The frame durations were as follows: 5 s × 1, 10 s × 5, 35 s × 1, 60 s × 2, 180 s × 1, and 300 s × 5. These sonograms were then reconstructed using a combination of two iterations of two-dimensional filtered back projection, followed by 18 iterations of the maximum a posteriori reconstruction algorithm. To analyze the PET images, region of interest (ROI) analysis was performed using PMOD software (PMOD Technologies LLC, Zurich, Switzerland). ROIs were delineated on all coronal brain images based on stereotactic coordinates. Specifically, ROIs were drawn for the hippocampus, cortex, and cerebellum. The non-displaceable binding potential, often utilized as an indicator of receptor binding density, was calculated by comparing the peak values of the specific binding curve (SUV hippocampus or cortex-SUV cerebellum) to the non-specific binding curve (SUV cerebellum) at the peak time. The cerebellum served as a reference region for this analysis [56].

### 4.4. Immunohistochemistry

The mice were anesthetized using Rompun (15 mg·kg^−1^) and Zoletil (8.3 mg·kg^−1^) before being perfused with saline solution. The brains were then extracted and fixed in 4%PFA at 4 °C for 24 h. Following fixation, the brains were incubated in a 30% sucrose solution at 4 °C for 72 h to ensure cryoprotection. Coronal sections with a thickness of 30 μm were obtained using a cryostat (Cryotome, Thermo Electron Corporation, Waltham, MA, USA) and stored at 4 °C. The opposite hemisphere of each brain was directly stored at −80 °C. To prepare the sections for immunohistochemistry, they were washed 3 times with PBS containing 0.4% Triton X-100 and then incubated in a blocking solution for 1 h at room temperature. Subsequently, the brain sections were incubated overnight at 4 °C with 1st antibodies targeting mGluR5 (Abcam, Cambridge, UK), Aβ (4G8, Bio Legend, San Diego, CA, USA) or p-tau (AT8, Thermo Scientific, Waltham, MA, USA). After washing the sections three times with PBS containing 0.4% Tween 20 (PBS-T), they were incubated with the suitable secondary antibody for 1 h at room temperature. Following another round of three washes, the brain tissues were mounted onto slides. Specimens were examined using a Nikon TS2-S-SM microscope (Nikon Microscopy, Nikon, Tokyo, Japan) equipped with a Nikon DS-Qi2 camera [23].

### 4.5. Western Blot Analysis

The cortex and hippocampus regions were homogenized using radioimmunoprecipitation assay (RIPA) buffer (150 mM NaCl, 1% NP-40, 0.5% sodium deoxycholate, 0.1% SDS, 50 mM Tris, pH 8.0) or Synaptic Protein Extraction Reagent (Thermo Scientific, Waltham, MA, USA) containing protease inhibitors (Calbiochem, San Diego, CA, USA) and a cocktail of phosphatase inhibitors (Sigma-Aldrich, St. Louis, MO, USA). The homogenates were centrifuged at 13,000× g rpm for 20 min or at 1200× g for 10 min at 4 °C, followed by an additional centrifugation at 15,000× g for 20 min. After lysate samples were quantified using a BSA assay (Bio-Rad Laboratories, Inc., Hercules, CA, USA), they were separated with SDS-PAGE and transferred to a PVDF membrane in a transfer buffer. The membrane blocked the 3% BSA in TBS-T for 1 h at room temperature and was then incubated with appropriate antibody mGluR5 (Abcam, Cambridge, UK), p-ERK (Cell Signaling, Denver, MA, USA), t-ERK (Cell Signaling, Denver, MA, USA), p-AKT (Cell Signaling, Denver, MA, USA), t-AKT (Cell Signaling, Denver, MA, USA), p-GSK-3β (Santa Cruz, Dallas, TX, USA), t-GSK-3β (Santa Cruz, Dallas, TX, USAsc-9166), AT8 (Thermo, MA, USA), AT180 (Thermo Scientific, Waltham, MA, USA), pS396 (Thermo Scientific, Waltham, MA, USA), and GAPDH (Santa Cruz, Dallas, TX, USA) overnight at 4 °C. After washing 3 times in TBS-T, the membrane was incubated with the suitable secondary antibody for 1 h. The protein band was detected using an Immobilon Western Chemiluminescent HRP Substrate (Millipore, Burlington, MA, USA) and BLUE X-ray film (AGFA, Mortsel, Belgium). The band quantification was performed using the Image J software (v1.4.3.67, NIH, Bethesda, MD, USA) [23].

### 4.6. ELISA (Aβ)

To quantify the levels of Aβ1-42 in the soluble fraction of the brain, an enzyme-linked immunosorbent assay (ELISA) was employed using the Aβ42 kit (Thermo Scientific, Waltham, MA, USA). The ELISA assay was conducted following the instructions provided by the manufacturer. The protein levels were calculated from a standard curve developed with specific optical density versus serial dilutions of a known concentration. Each standard and protein progressed in duplicate, and the results were averaged [23].

### 4.7. Statistical Analysis

All values are expressed as the mean ± SEM. All statistical analyses were performed using GraphPad Prism 8 software (GraphPad, La Jolla, CA, USA). For the MWM training test, two-way repeated-measures ANOVA with Bonferroni multiple comparison corrections was used to compare the escape latency across 4 days of continuous hidden platform trials. The MWM probe test was analyzed with one-way ANOVA followed by Tukey’s multiple comparisons test. Data obtained from the anxiety- and depressive-like behavior tests and immune assays were analyzed using the unpaired t-test. Correlations were assessed using the nonparametric Spearman’s rank correlation test. Graphs show regression lines with 95% confidence intervals. A value of *p* < 0.05 was considered statistically significant (* *p* < 0.05, ** *p* < 0.01, *** *p* < 0.001, and **** *p* < 0.0001) [57]).

## Figures and Tables

**Figure 1 ijms-24-13010-f001:**
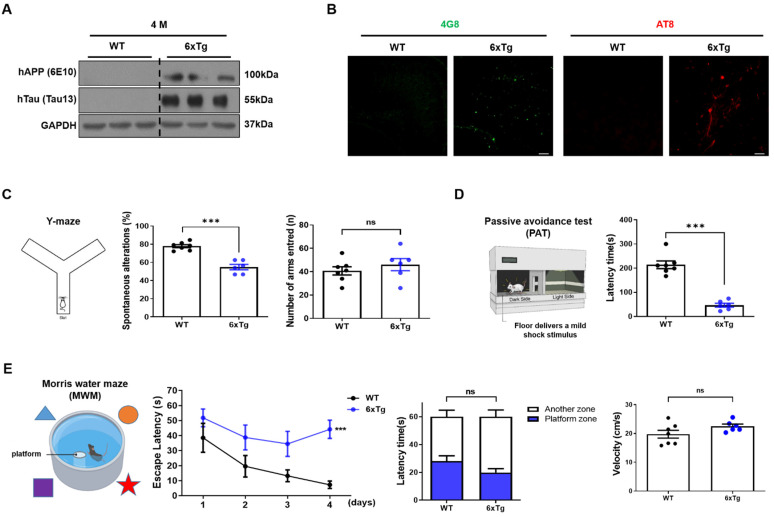
Formation of Aβ and p-tau and cognitive impairment in 6xTg mice. (**A**) Memory deficits in 4-month-old 6xTg mice. The human APP and tau were measured in the cortex of mice (*n* = 3 per group). (**B**) The immunohistochemistry (IHC) image showed the Aβ and p-tau staining using 4G8 and AT8 antibodies in the brain of mice (*n* = 3 per group 100 μm). Cognitive impairment was observed using (**C**) Y-maze test (*n* = 6–7 per group), (**D**) Passive avoidance test (*n* = 6–7 per group), and (**E**) Morris water maze (*n* = 6–7 per group) in 6xTg mice. Data are presented as means ± SEM. ns = non significant, *** *p* < 0.001 vs. WT. Statistical significance between the two groups was determined using the Student’s *t*-test and for MWM, a two-way repeated-measures ANOVA followed by a Bonferroni multiple comparisons correction was used to compare the escape latency in 4 days of continuous hidden platform trials.

**Figure 2 ijms-24-13010-f002:**
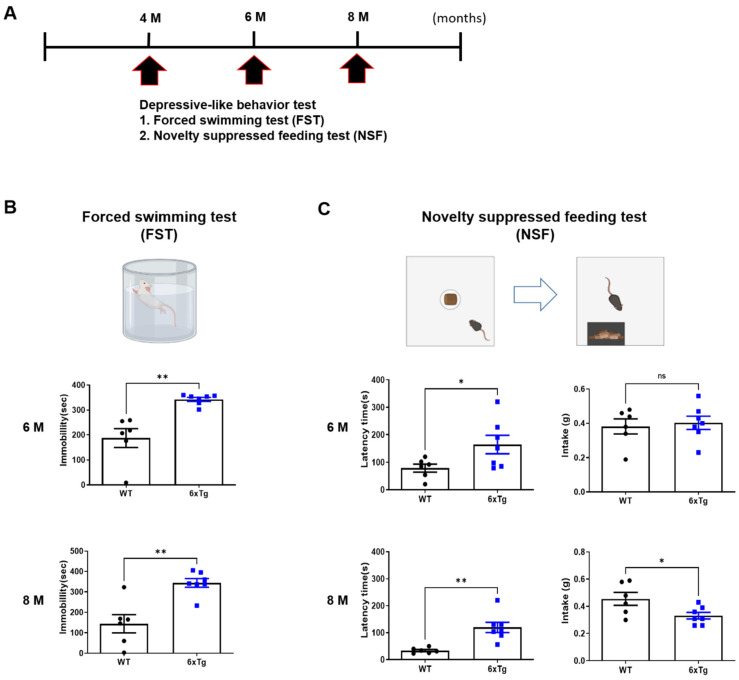
Depressive-like behaviors in 6xTg mice. (**A**) Depressive-like behavior experiments, including forced swimming test and novelty suppressed feeding test, were conducted on WT and 6xTg mice aged 6 and 8 months. (**B**) The forced swimming test (FST) and (**C**) novelty-suppressed feeding test (NSF) were carried out to assess and confirm the presence of depressive-like behaviors. Data are presented as means ± SEM (*n* = 6–7 per group). * *p* < 0.05 and ** *p* < 0.01 vs. WT. Statistical significance between the two groups was determined using the Student’s *t*-test.

**Figure 3 ijms-24-13010-f003:**
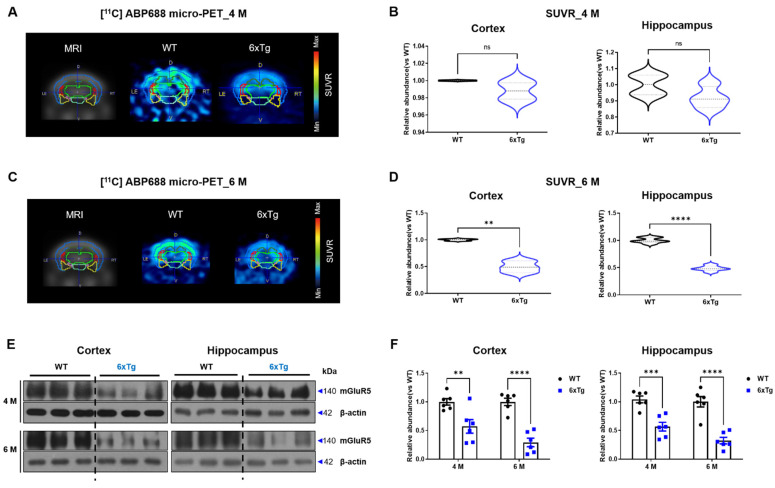
Reduction in mGluR5 density in the brains of 6xTg mice. (**A**,**C**) Representative images of [11C]-ABP688 in the brain of 4- and 6-month-old WT and 6xTg mice using micro-PET. (**B**,**D**) Quantitative analysis of ROI including cortex and hippocampus (*n* = 3 per group each month). (**E**) Representative Western blot images of mGluR5 protein levels in cortex and hippocampus of 4- and 6-month-old WT and 6xTg mice. (**F**) Quantitative analysis of mGluR5 expressions in the cortex and hippocampus (*n* = 6 per group each month). Data are presented as means ± SEM. ** *p* < 0.01, *** *p* < 0.001 and **** *p* < 0.0001 vs. WT. Statistical significance between the two groups was determined using the Student’s *t*-test.

**Figure 4 ijms-24-13010-f004:**
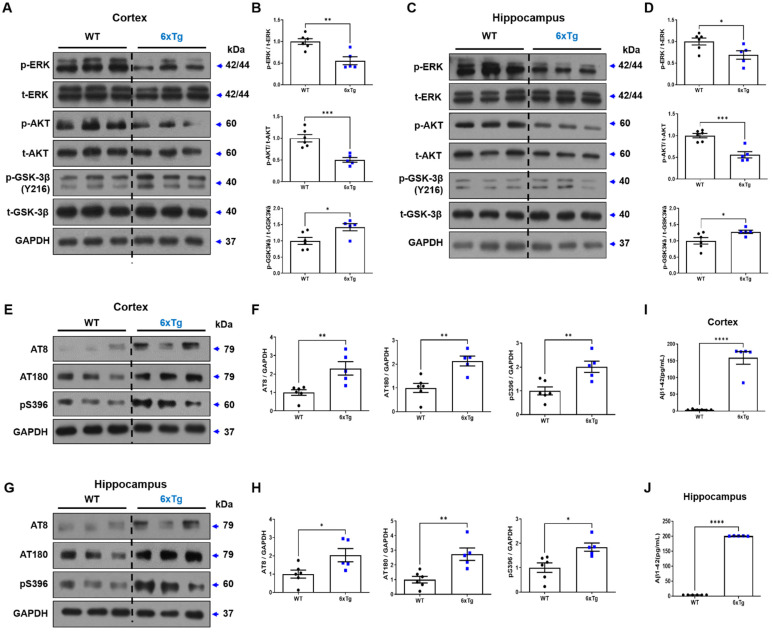
Alteration of ERK/AKT/GSK-3β signaling pathway in the brains of 6xTg mice. (**A**) Representative Western blot images of p-ERK/t-ERK/p-AKT/t-ERK/p-GSK-3β/t-GSK-3β protein levels in the cortex of 8-month-old WT and 6xTg brains. (**B**) Quantitative analysis of p-ERK/t-ERK/p-AKT/t-ERK/p-GSK-3β/t-GSK-3β expressions in the cortex. (**C**) Representative Western blot images of p-ERK/t-ERK/p-AKT/t-ERK/p-GSK-3β/t-GSK-3β protein levels in the hippocampus. (**D**) Quantitative analysis of p-ERK/t-ERK/p-AKT/t-ERK/p-GSK-3β/t-GSK-3β expressions in the hippocampus. (**E**) Representative Western blot images of phosphorylated tau proteins in the cortex using AT8/AT180/pS396 antibodies. (**F**) Quantitative analysis of phosphorylated tau proteins in the cortex. (**G**) Representative Western blot images of phosphorylated tau proteins in the hippocampus using AT8/AT180/pS396 antibodies. (**H**) Quantitative analysis of phosphorylated tau proteins in the hippocampus. (**I**,**J**) Aβ42 protein levels in the cortex and hippocampus of mice were quantified using ELISA. All data are given as means ± SEM (*n* = 6 per group). * *p* < 0.05, ** *p* < 0.01, *** *p* < 0.001, and **** *p* < 0.0001 vs. WT. Statistical significance between the two groups was determined using the Student’s *t*-test.

**Figure 5 ijms-24-13010-f005:**
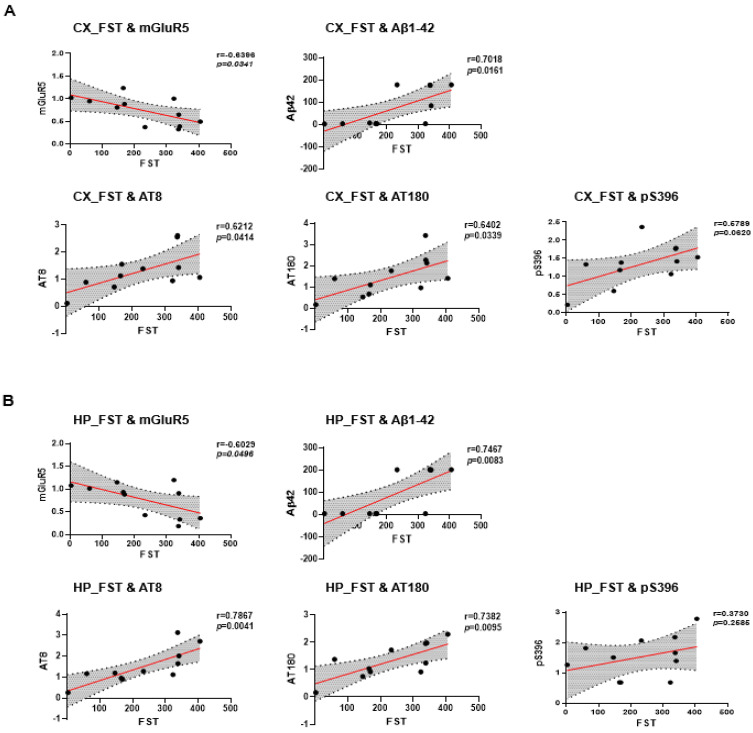
Correlations between depressive-like behavior and AD pathology in 6xTg mice. (**A**) The correlation between FST (depressive-like behavior) scores mGluR5, Aβ42, and p-tau (AD pathology) levels in the cortex of both WT and 6xTg mice. (**B**) The correlation between FST scores mGluR5, Aβ42, and p-tau levels in the hippocampus of both WT and 6xTg. Data are presented as means ± SEM (*n* = 5 per group). *p* < 0.05 compared to the WT group with one-way ANOVA and post hoc Dunn’s multiple comparison test. Correlations were assessed using the nonparametric Spearman’s rank correlation test. Graphs show regression lines with 95% confidence intervals.

**Figure 6 ijms-24-13010-f006:**
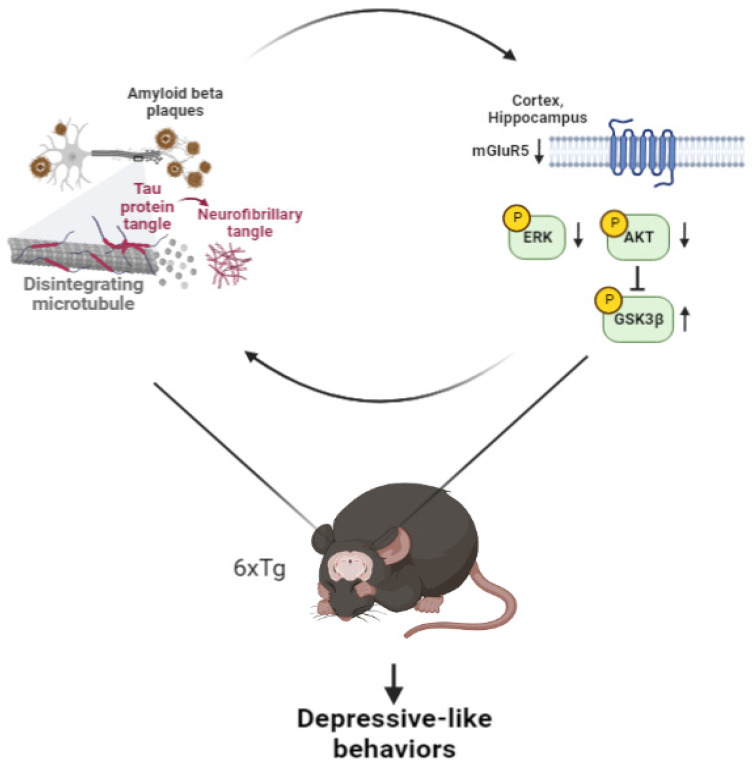
Summary picture. Depression-like behavior correlated with mGlur5 expression or AD pathologies such as Aβ and tau pathologies. p-ERK causes Aβ deposition, and p-AKT increases pGSK-3B by inhibiting p-tau and increasing it. As a result, it affects depression-like behavior in 6xTg mice, which show increased levels of Aβ and p-tau compared to WT mice.

## Data Availability

The data supporting the findings of this study are available from the corresponding author upon reasonable request.

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
