# Peer review of "Depressive-like Behaviors Induced by mGluR5 Reduction in 6xTg in Mouse Model of Alzheimer’s Disease"

_ijms, 2023, doi:10.3390/ijms241613010_

Round 1
Reviewer 1 Report
The research wants to investigate further (as other groups in the field) if there is a correlation between the expression of mGluR receptor and AD. In their study, they used a transgenic 6xTg model assessing behavior-like symptoms such as depression and appearance of inclusion/aggregates of tau and abeta.
It is relevant but not original. It supports studies of other groups that there is a possible correlation of mGluR5 expression and AD. The authors should include females in their mouse studies, and can be further improved with more detailed investigation of the mechanism of how mGluR5 reduction could affect depression in mice, and how the increase in abeta or tau phosphorylation has an effect on mice depression-like symptoms. The tables and figures are ok with the significance on the figures. For non-significant data, maybe they can add non-significant in the figures.Author Response
Please see the attachment

Reviewer 2 Report
In the present study by Kim et al., the authors showed that the metabotropic glutamic receptor subtype 5 (mGluR5) was upregulated in 6xTg Alzheimer disease (AD) model mice and was involved in the depressive-like behaviors in these mice. Because it has been known that many AD patients demonstrate neuropsychiatric symptoms including depressive-like behaviors, the current results would be of clinical importance. The manuscript may be accepted for publication after a revision.
1. In Fig. 1, the authors only showed Abeta staining and phosphorylation of tau. Because Figs 1C-E show cognitive impairment in 6xTg mice, synaptic and neuronal loss also need to be shown.
2. In Fig.4, the authors said increased phosphorylation of GSK3beta at Tyr216, but decreased phosphorylation of Akt. Akt phosphorylates the Ser at the N-terminus of GSK3, which can override the Tyr phosphorylation. Line 202-203 on page 6, “The increase in the p-GSK-3β/t-GSK-3β ratio indicates elevated enzyme activity and may contribute to abnormal tau phosphorylation and amyloid production.” needs a reference. GSK3alpha is reportedly involved in increased Abeta production. I disagree with that “The increase in the p-GSK-3β/t-GSK-3β ratio indicates elevated enzyme activity and … may contribute to amyloid production”, because the authors compared Abeta production and pTau between WT mice and 6xTg mice. WT mice do not carry mutant human APP and mutant human tau. The increased Abeta and pTau must have been arisen from the transgenes. Whether the depressive-like behaviors and the reduction in mGluR5 levels in 6xTg mice were caused by Abeta pathology or tau pathology is also unclear. Comparing the 6xTg mice and P301L tau tg mice will be helpful.
3. Fig. 6 is incorrect. No evidence of “neurotoxicity” and mGluR5 reduction has been provided, and no evidence that reduced mGluR5 resulted in Abeta deposition and NFT formation has been provided. The authors only showed reduced mGluR5 and depressive-like behaviors in 6xTg mice which were by the transgenes. It is not clear whether depressive-like behaviors occur after the occurrence of AD pathologies in the late-onset AD. This is important because 6xTg mice carry familial mutants of AD-related genes, but more than 90% of AD cases are late-on-set. Use of human samples will strengthen the current findings.
Minor
1. Please provide the information of the transgenes of 6xTg mice. Information and references of the responsible kinases for the induction of the pTau antibody epitopes are also helpful.
2. On line 90, “passive assistance test (PAT)” will be “passive avoidance test (PAT)”.
Round 2
Reviewer 2 Report
The revised version has been sufficiently improved.